# Comparing the Use of Red-Edge and Near-Infrared Wavelength Ranges for Detecting Submerged Kelp Canopy

Brian Timmer [1],*, Luba Y. Reshitnyk [2], Margot Hessing-Lewis [2], Francis Juanes [3] and Maycira Costa [1]

1    Spectral Lab, University of Victoria, 3800 Finnerty Road, Victoria, BC V8P 5C2, Canada; maycira@uvic.ca
2    Hakai Institute, Heriot Bay, BC V0P 1H0, Canada; luba.reshitnyk@hakai.org (L.Y.R.); margot@hakai.org (M.H.-L.)
3    Department of Biology, University of Victoria, 3800 Finnerty Road, Victoria, BC V8P 5C2, Canada; juanes@uvic.ca
*    Correspondence: briantimmer@uvic.ca

**Abstract:** Kelp forests are commonly classified within remote sensing imagery by contrasting the high reflectance in the near-infrared spectral region of kelp canopy floating at the surface with the low reflectance in the same spectral region of water. However, kelp canopy is often submerged below the surface of the water, making it important to understand the effects of kelp submersion on the above-water reflectance of kelp, and the depth to which kelp can be detected, in order to reduce uncertainties around the kelp canopy area when mapping kelp. Here, we characterized changes to the above-water spectra of *Nereocystis luetkeana* (Bull kelp) as different canopy structures (bulb and blades) were submerged in water from the surface to 100 cm in 10 cm increments, while collecting above-water hyperspectral measurements with a spectroradiometer (325–1075 nm). The hyperspectral data were simulated into the multispectral bandwidths of the WorldView-3 satellite and the Micasense RedEdge-MX unoccupied aerial vehicle sensors and vegetation indices were calculated to compare detection limits of kelp with a focus on differences between red edge and near infrared indices. For kelp on the surface, near-infrared reflectance was higher than red-edge reflectance. Once submerged, the kelp spectra showed two narrow reflectance peaks in the red-edge and near-infrared wavelength ranges, and the red-edge peak was consistently higher than the near-infrared peak. As a result, kelp was detected deeper with vegetation indices calculated with a red-edge band versus those calculated with a near infrared band. Our results show that using red-edge bands increased detection of submerged kelp canopy, which may be beneficial for estimating kelp surface-canopy area and biomass.

**Keywords:** kelp; hyperspectral; multispectral; red-edge; near-infrared; satellite; unoccupied aerial vehicle; remote sensing

## 1. Introduction

Kelp forests are highly productive three-dimensional coastal marine habitats [1,2] that provide a number of environmental services and contribute substantial economic value to coastal communities globally [3]. In the northeast Pacific, the two dominant surface-canopy forming kelp species, *Nereocystis luetkeana* and *Macrocystis pyrifera* [4], stabilize shorelines via wave dampening [5,6], support economically important fisheries [7,8], and are commercially harvested for various purposes [9,10]. However, both kelp species are subject to high spatial and temporal variability, correlated with biotic and abiotic drivers of change [11,12]. As such, resource managers are incentivized to monitor the status of these kelp forests, and the corollary effects of the ecosystem services they provide [10,12,13], a task that has been facilitated by remote sensing since the mid-20th century [12,14].

Generally, the remote sensing of surface-canopy forming kelp forests aims to detect the portion of the kelp that forms a canopy, floating at the water's surface; using sensors

aboard Earth Observation Satellites (EOS) [15,16], piloted aircraft [10,12], and Uncrewed Aerial Vehicles (UAVs) [17]. In order to use the data provided by remote sensing platforms effectively, it is crucial to understand factors that influence the spectral signature of kelp canopy in water [16]. Floating kelp canopy has high reflectance in the near-infrared wavelength range (NIR) (700–1000 nm), which contrasts with the high NIR absorption by the surrounding water, allowing for binary classification of floating kelp canopy and water within an image [14]. However, there are numerous considerations (e.g., sun glint, bathymetry, turbidity; see [13,16]) that can reduce the separability between the spectral values of kelp canopy and water. One crucial factor that can affect the ability to detect kelp canopy is the submersion of the canopy by tides and associated tidal currents, which can dampen the NIR reflectance of kelp and lead to potential errors when estimating kelp area or biomass [17,18].

In an attempt to minimize classification errors associated with kelp submergence, remote sensing imagery is often acquired at low tides during the peak growing season (mid-late summer) when the majority of the kelp canopy is floating at the water's surface [10,18,19]. However, there are multiple reasons why a remote sensor may also want to detect the submerged portion of the kelp canopy. For example, the northeast Pacific coastline often experiences non-ideal weather conditions for remote sensing data acquisition, leading to imagery being opportunistically collected at higher than ideal tidal heights when more kelp canopy is more likely to be submerged compared to ideal low tide conditions [13,16]. Further, the fixed rate of EOS orbits may result in some regions only having imagery available during high tides even if acquisition conditions are otherwise ideal [13]. Even if remote sensing imagery is captured during ideal tide and weather conditions, portions of kelp canopy may also be continuously submerged depending on the species being targeted. Specifically, if a remote sensor is targeting detection of *Nereocystis luetkeana* (hereafter, *Nereocystis*) surface canopy, one has to consider the two distinct structures with varying buoyancy, the bulb and blades. The bulb is a roughly cylindrical gas-filled structure that floats on the surface of the water and is anchored to the sea floor by a stipe and holdfast [9]. The blades are long thin structures that trail from the end of the bulb, often with many individuals around four meters long per bulb [9]. The blades are not buoyant and are likely to remain submerged below the water's surface regardless of tidal height [16,20]. In addition, floating portions of kelp canopy may be periodically submerged in areas with especially strong currents [20]. Therefore, it is important to understand how submersion of kelp canopy affects the reflectance in the NIR range, as well as whether certain spectral features may allow for higher detectability of kelp when collecting remote sensing imagery from different platforms.

In the past, the red-edge (RE) spectral region (670–750 nm), which includes a range of the shortest NIR wavelengths, has traditionally been used to determine health characteristics of terrestrial plants [21]. However, these wavelength ranges also penetrate deeper into the water column than longer NIR wavelength ranges [22], resulting in the potential for higher above water reflectance in the RE than the longer NIR for submerged vegetation [23–26]. Therefore, given the spectral similarities between kelps and other types of vegetation, it is reasonable to assume that the RE wavelength range may also be beneficial for detecting submerged kelp canopy. Hereafter, the term NIR will refer to only the longer wavelength range above 751 nm, to avoid confusion with the NIR wavelength range that overlaps the RE wavelength range.

To date, there have been no direct comparisons of the ability to detect submerged kelp when using RE or NIR wavelength ranges. Additionally, while the submersion kelp canopy due to tides and currents is well documented using various sensors with different spatial and spectral resolutions [11,17–20,27], there has been no characterization of the changes to the above water spectra of kelp as the canopy is submerged, nor any investigation of the band combinations used in vegetation indices in relationship to accurate detection of the submerged kelp canopy. With this is mind, our goal was to characterize changes to above-water reflectance of different *Nereocystis* canopy structures as they were submerged and to

relate those changes to depth detection limits. To accomplish this goal, we performed (1) an experiment that documents the effects of kelp submersion on the above-water hyperspectral reflectance of both *Nereocystis* bulb and blade structures. We also compared (2) the detection limits of submerged kelp using RE and NIR vegetation indices, which were calculated from the simulated multispectral bands of high spatial-resolution air- and space-borne sensors.

## 2. Materials and Methods

### 2.1. Spectral Data Acquisition and Processing

The kelp submergence experiment took place on a marina dock in Victoria BC, on a sunny, cloudless day in September 2020. The Secchi depth during the time of the experiment was 7.5 m, showing relatively clear water, similar to general conditions for the coastal waters of the Salish Sea at the same time of year with low influence from riverine discharge, and low levels of total suspended-matter, chlorophyll-a, and colored dissolved organic matter present in the water column [28]. While the ranges of both Nereocystis and Macrocystis overlap on the British Columbia coast, only Nereocystis is found around the southern tip of Vancouver Island where this study occurred. The location and timing of the experiment allowed for the control of four criteria that we required: (1) controlled sea-state; with the dock acting as a shelter from any slight breezes, thereby minimizing variability in glint or light refraction due to ripples or waves on the water [29]; (2) platform stability, which minimized the potential errors during spectral acquisition due to the movement of both kelp and sensor that might occur in situ from a boat; (3) maintained environmental conditions expected in situ during peak biomass for local kelp, such as the inherent optical properties of water and optical constituents within the water column that would be difficult to reproduce in vitro; (4) a water depth (12 m) greater than the Secchi depth to minimize the influence of substrate reflectance on the above-water reflectance signal [30,31].

The experiment consisted of four separate trials. For each trial, a sample of *Nereocystis* was attached to a black frame made of high-density polyethylene (a plastic with low reflectance across the visual and near-infrared wavelength ranges), which was submerged from the surface to 100 cm in 10 cm increments on the sunlit side of the dock (Figure 1). Before each trial, radiance measurement of a Spectralon white-reference panel ($L_{spec}(\lambda)$) and an internal dark-current reading were taken to calculate reflectance (Table 1; Equation (1)) and reduce noise in the spectral data [32]. During each trial, ten individual above-water hyperspectral radiance measurements ($L_T(\lambda)$) of kelp were collected at each incremental depth. Two of the four trials used the *Nereocystis* bulb, and two trials used the *Nereocystis* blades. Therefore, in total, 20 measurements of $L_T(\lambda)$ were collected for each kelp structure (bulb or blades) at each depth. After each trial, 10 radiance measurements were taken of the sky ($L_{sky}(\lambda)$) to be used in sky glint corrections [33]. Additionally, a total of 60 $L_T(\lambda)$ measurements were taken of water with no kelp within the field of view as a baseline for comparison with submerged kelp.

**Table 1.** Spectral parameters used to calculate above-water reflectance, as per Equation (1). All spectral measurements were collected using a calibrated ASD Fieldspec Handheld2 spectroradiometer with a one-degree fore optic (full viewing-angle), which detects a wavelength range from 325–1075 nm at 1 nm increments.

| Symbol | Name | Units | Angle from Nadir | Sun-Sensor Azimuthal Angle |
|---|---|---|---|---|
| $\lambda$ | Wavelength | nm | - | - |
| $L_T$ | Above-water radiance | $\mu W\ cm^{-2} sr^{-1} nm^{-1}$ | 5° | 135° |
| $L_{spec}$ | White panel radiance | $\mu W\ cm^{-2} sr^{-1} nm^{-1}$ | 5° | 135° |
| $L_{sky}$ | Sky radiance | $\mu W\ cm^{-2} sr^{-1} nm^{-1}$ | 175° | 135° |
| $\rho'$ | Proportionality factor | - | - | - |

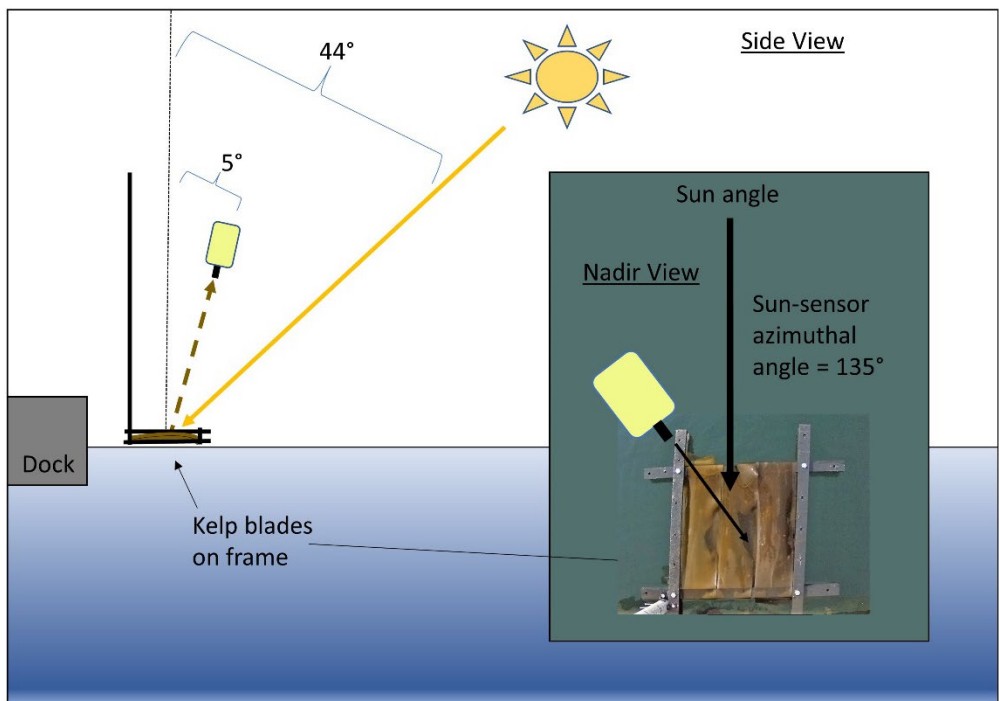

**Figure 1.** Side view of submergence experiment showing the geometry of acquisition for spectroradiometer and angle of zenith for the sun. Inset shows nadir view of the experiment with the azimuthal angle between spectroradiometer and sun and kelp blades inside the black frame. Diagrams are not to scale.

The solar elevation angle during the experiment was 46°, which ensured sun-glint did not contaminate the spectra based on our geometry of acquisition [33,34]. $L_T(\lambda)$ measurements were taken at 5° from a nadir viewing angle to avoid reflection of the white spectroradiometer in the field of view on the water surface, and a sensor-sun azimuthal angle of 135° was used to minimize specular reflection in the field of view (FOV) [33]. $L_{sky}(\lambda)$ measurements were taken at 5° from zenith at the same azimuthal angle as $L_T(\lambda)$. The spectroradiometer was held one meter above water, giving a footprint ranging from about 1.6 cm at the surface to 3.8 cm when the target was 100 cm deep. This small footprint was meant to ensure that the $L_T(\lambda)$ measurements contained 100% kelp, avoiding mixed pixel considerations [35].

$$R(\lambda)_{0+}(\%) = \left( \frac{(L_T(\lambda))}{(L_{spec}(\lambda))} - \frac{\left( \rho' \cdot L_{sky}(\lambda) \right)}{(L_{spec}(\lambda))} \right) \times 100 \qquad (1)$$

Here, $\rho'$ was the proportionality factor of 0.0211, which relates the radiance measured directly from the sky to the estimated amount of sky radiance reflected off the sea surface based on wind, cloud cover, and geometry of acquisition [33]. $R(\lambda)_{0+}(\%)$ for kelp at the surface (0 cm) was not subjected to the sky glint correction. Hereafter, $R(\lambda)_{0+}(\%)$ values for kelp on the surface, submerged kelp, and water with no kelp are referred to as $R_{0+}$ for brevity.

The $R_{0+}$ spectra were first smoothed using a mean filter with a window of 5 nm to reduce noise while maintaining spectral features, and all spectra were then manually inspected for quality control. All bulb spectra were highly consistent, however, some blade spectra showed deviations in both the blue-green and NIR regions; likely due to water movement between blades as kelp was submerged, causing opened gaps in the "canopy" of blades attached to the platform. These spectra likely did not contain 100% blades within the field of view and were therefore removed from further analysis (Table 2). Despite the

removal of some blade spectra, the smallest sample size at any depth after quality control was at 90 cm with $n = 10$ spectral samples. Therefore, we do not expect that these removals biased the results of this study.

**Table 2.** Total number of each class of spectra before and after quality control was performed.

| Spectral Sample Type | Samples Collected | Samples Removed during Quality Control | Samples Used in Analysis |
|---|---|---|---|
| Bulbs (surface-100 cm) | 220 | 0 | 220 |
| Blades (surface-100 cm) | 220 | 51 | 169 |
| Water | 60 | 0 | 60 |
| Sky | 40 | 0 | 40 |

### 2.2. Simulation of Micasense and WorldView Band $R_{0+}$ and Indices

After sky-glint correction, smoothing, and quality control of the spectra, $R_{0+}$ measurements were simulated into bands of the WorldView-3 ($R_{0+WV3}$) and the Micasense RedEdge-MX ($R_{0+MSRE}$) sensors [36,37]. These sensors were chosen because both have a relatively high spatial resolution (WV3: 1.84 m; MSRE: ~1–10 cm), which is ideal for mapping kelp canopy in nearshore regions where it is likely to be submerged by tides and currents [16]. The $R_{0+}$ at the bands of these sensors were simulated using Gaussian functions to estimate the sensor's spectral response for each band, based on full-width half maximum values of each sensor's band (Figure 2; Table 3). For a direct comparison, only the VNIR bands shared by both sensors were used for simulations.

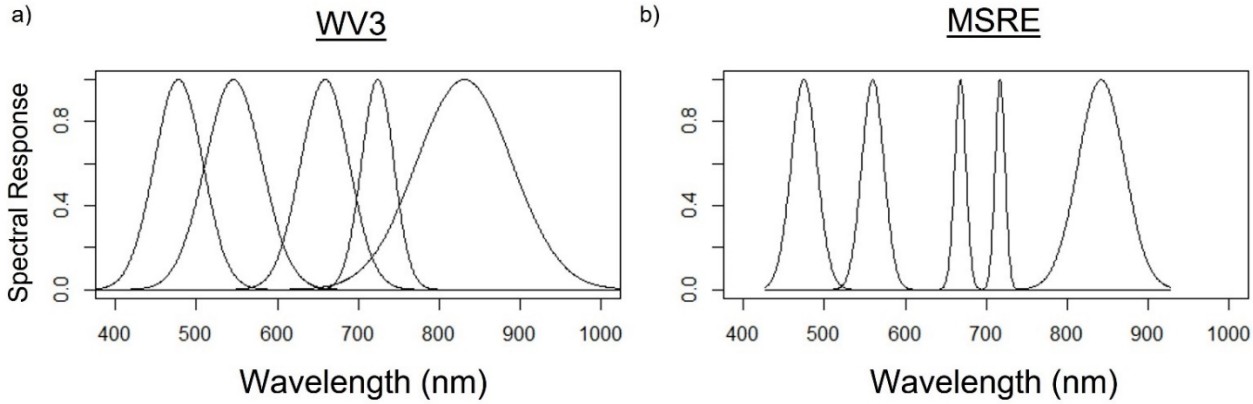

**Figure 2.** Relative spectral responses at each band according to Gaussian functions were used to simulate the shared bands of (**a**) WorldView-3 (WV-3) earth observation satellite and (**b**) Micasense RedEdge-MX (MSRE) uncrewed aerial vehicle sensors—from left to right: blue, green, red, red-edge, and near-infrared band locations are shown.

**Table 3.** The effective bandwidths of the overlapping bands for both WorldView-3 (WV3) and Micasense RedEdge-MX (MSRE) sensors.

| Band | WV3 | MSRE |
|---|---|---|
| Blue | 445–517 nm | 459–491 nm |
| Green | 507–586 nm | 546.5–573.5 nm |
| Red | 626–696 nm | 661–675 nm |
| Red-edge | 698–749 nm | 711–723 nm |
| Near-infrared | 765–899 nm | 813.5–870.5 nm |

### 2.3. Normalized Vegetation Indices

Once the hyperspectral data were simulated into the respective sensor bands, the $R_{0+}$ at these bands were used to calculate normalized vegetation indices ($VI_n$; Equation (2)), which are commonly used to enhance spectral features of interest and reduce sensitivity to environmental influences within remote sensing imagery [38,39]. We tested several band combinations for $VI_n$ as different band combinations may increase or decrease the separability between kelp and water in an image [40].

$$VI_n = \frac{band\ 2 - band\ 1}{band\ 2 + band\ 1} \tag{2}$$

Because naming conventions for different $VI_n$ combinations are not ubiquitous across published literature, here, we referred to each $VI_n$ as the order in which bands appeared in the numerator of the $VI_n$ equation, separated by an underscore (Table 4).

**Table 4.** Vegetation indices calculated from simulated multispectral data.

| Vegetation Index ($VI_n$) | $VI_n$ Equation |
|:---:|:---:|
| RE_R | $\frac{RE-red}{RE+red}$ |
| RE_G | $\frac{RE-green}{RE+green}$ |
| RE_B | $\frac{RE-blue}{RE+blue}$ |
| NIR_R | $\frac{NIR-red}{NIR+red}$ |
| NIR_G | $\frac{NIR-green}{NIR+green}$ |
| NIR_B | $\frac{NIR-blue}{NIR+blue}$ |

One of the most commonly used $VI_n$ for kelp mapping is NIR_R, which was originally used to detect terrestrial vegetation because of the high NIR and low red signal [38], but has since been used for kelp canopy detection due to the similar spectral characteristics between kelp canopy and terrestrial vegetation [14]. More recently, NIR_R has been positively correlated with both the areal extent and biomass of kelp canopy [15,18,19]. However, various other combinations of visible and NIR bands have been used for kelp canopy detection with multispectral sensors. For instance, Schroeder et al. (2019b) used NIR_R and NIR_G for kelp detection with the WorldView-2 imagery. The NIR_G combination may be more accurate for detecting a wide range of chlorophyll levels [41] and has generally been found comparable with NIR_R in the detection of both floating and submerged vegetation [26]. Stekoll et al. (2006) found that NIR_B and NIR_G both provided higher kelp canopy and water separability in aerial imagery than NIR_R. Further, recent comparisons with multispectral UAV and satellite imagery have shown that RE indices can improve separability of Macrocystis canopy and water when compared with NIR based indices [17,42], although this improvement was not specifically attributed to improved detection of submerged portions of the kelp canopy in either study.

Here, we compared the statistical differences in NIR and RE-based $VI_n$ values. $R_{0+MSRE}$ and $R_{0+WV3}$ bands were used to calculate NIR_B, NIR_G, NIR_R, and RE_B, RE_G, and RE_R for both bulb and blades separately, for each depth. The statistical analysis was comprised of (i) $VI_n$ values compared with one another at each depth from the surface to 100 cm, and (ii) $VI_n$ values for water (with no kelp) compared to one another. First, the dataset was tested for normality, and while quantile–quantile plots suggested reasonable normality of the data distributions, Levene's test showed nearly all groupings for comparison displayed heterogeneity in variance. Therefore a non-parametric test was used in the analysis [43]. The Welch's ANOVA test was used to determine whether significant differences between $VI_n$ existed at each depth, and the Games–Howell post hoc test was used to determine which indices were significantly different from one another [44,45]. As part of the analysis, we focused on the statistical results comparing the RE and NIR counterpart indices only (e.g., NIR_R & RE_R, or NIR_B & RE_B) at each depth.

*2.4. Threshold Selection and Depth Limits for Kelp Detection*

Once a $VI_n$ has been selected for classifying kelp in remote sensing imagery, a $VI_n$ value is then chosen as a threshold to classify the kelp and water within the imagery. For example, Cavanaugh et al. (2010) selected a threshold based on the 99.98th percentile highest NIR_R value from a histogram of known 'deep water' pixels, and Nijland et al. (2019) determined a NIR_R value of 0.05 to be a reasonable threshold by comparing pixel values of sparse kelp and open water. Since the $R_{0+}$ values of water vary spatially and temporally according to optical constituents and inherent optical properties of water, as well as the characteristics of local substrate and bathymetry [28,46,47], these thresholds are often 'dynamic', and are therefore determined on an image-by-image basis. For satellite or airborne imagery covering a large regional scale, it may even be appropriate to select multiple thresholds across different regions within an image.

We determined a dynamic threshold for each $VI_n$ based on the maximum $VI_n$ value measured for water during the experiment following Cavanaugh et al. (2010). The depth where the mean $VI_n$ value of submerged kelp dropped below the dynamic threshold value was considered the depth where kelp was spectrally indistinguishable from water. Since our experiment was conducted under ideal conditions (flat calm water, full sun, etc.) the dynamic thresholds were all negative values and the maximum depth of detection using these thresholds likely overstate the potential depths for kelp detection in actual remote sensing imagery. Therefore, we also used a second $VI_n$ threshold of zero, based on the theoretical spectral properties of kelp within an individual pixel that contains 100% kelp. For example, within a pixel, if the $R_{0+}$ value of band 2 (RE or NIR) equals the $R_{0+}$ value as band 1 (the visible band), the numerator in the $VI_n$ equation (Equation (2)), and therefore the overall $VI_n$ value for that pixel, equals zero. This conservative threshold is closer to the values of 0.05 and 0.003 determined from remote sensing imagery by Nijland et al. (2019) and Mora-Soto et al. (2020), respectively.

Depth detection limits were reported to the nearest 10 cm depth on the shallow side of the threshold because the kelp was submerged in 10 cm intervals. To determine whether the detectable kelp (values above the threshold) and non-detectable kelp (values below the threshold) were statistically separable, the means for kelp measurements immediately above and below the threshold were compared for significant differences using Welch's *t*-test [48].

## 3. Results

Here, we present the spectral characteristics of *Nereocystis* bulbs and blades as they are each submerged from the surface to 100 cm, as well as the changes seen in the hyperspectral data when they are simulated into multispectral sensor bandwidths. Next, we show $VI_n$ comparisons for kelp, focusing on comparing the RE and NIR counterpart indices (e.g., NIR_R & RE_R, or NIR_B & RE_B) at each depth, and finally, we present the depth detection limits for each $VI_n$ as determined by both dynamic and conservative thresholds.

*3.1. Spectral Characteristics of Surface and Submerged Kelp*

Overall, the $R_{0+}$ of both *Nereocystis* bulbs and blades showed similar placement of spectral features, however, the magnitude of reflectance at these features was different (Figure 3a,b). For *Nereocystis*, spectral features in the visible wavelength ranges are largely due to absorption by a combination of chlorophyll-a, chlorophyll-c, and fucoxanthin pigments, which are characteristic pigments of bull kelp, as well as other kelp species [49,50]. Accordingly, here we saw a broad absorption feature in the 400–550 nm range and narrower absorption features around 633 and 675 nm for both bulbs and blades at the surface. These absorption features resulted in reflectance peaks at 575, 600, and 645 nm for both bulbs and blades (Figure 3a,b, insets). In the NIR region, broad reflectance peaks were detected from 690 nm (RE) to 900 nm (NIR) (Figure 3a,b, insets) and small, narrow peaks centered at 761 nm were observed (Figure 4a,b).

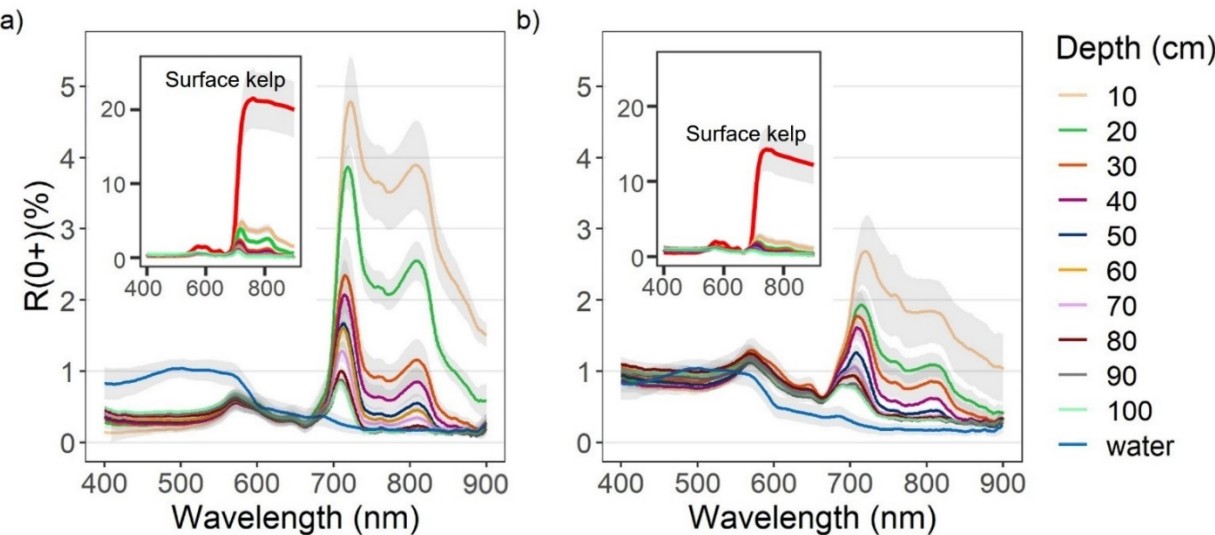

**Figure 3.** Reflectance values ($R_{0+}$) between 400–900 nm (mean +/− sd) of water with (**a**) *Nereocystis* bulbs, and (**b**) *Nereocystis* blades, at incremental depths below water surface. The inset plots contain spectra of bulbs and blades on the surface compared to the same spectra of submerged bulb and blades as in the main plots, for the purpose of showing the difference in magnitude.

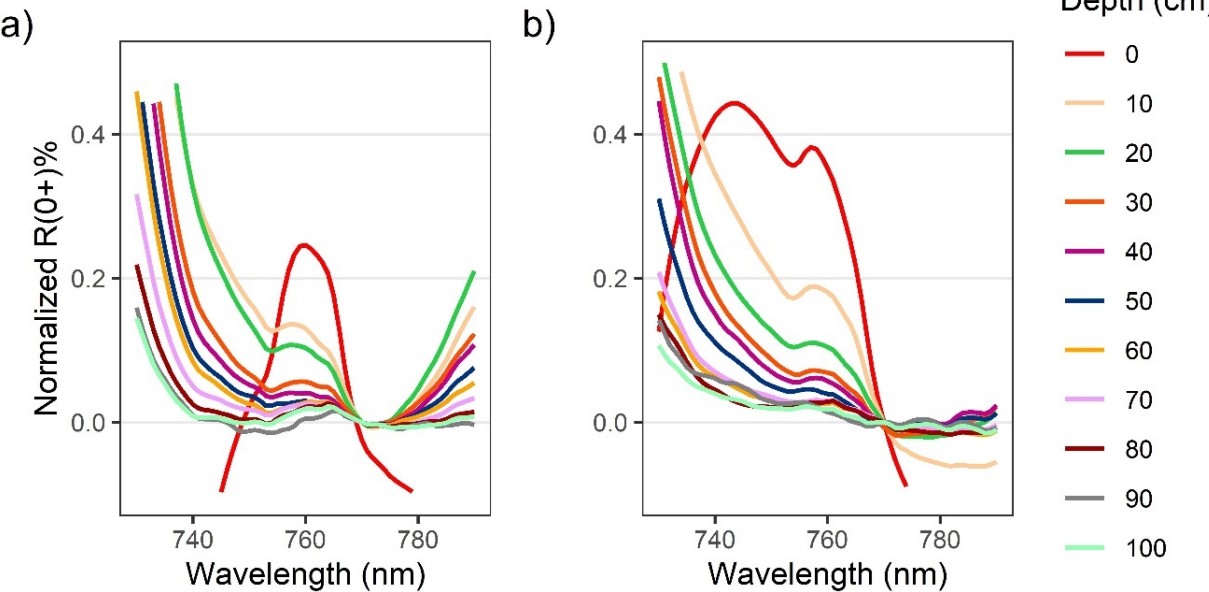

**Figure 4.** Zoomed in plot showing solar-induced chlorophyll fluorescence (SICF) peaks centered at 761 nm for above-water $R_{0+}$ for *Nereocystis* bulbs (**a**) and blades (**b**) at incremental depths below the water surface. Spectra are normalized at 770 nm to show relative changes to the shape of the SICF peak with submergence.

When kelp structures were submerged, the influence of the water and its constituents on the $R_{0+}$ signal increased with submersion for both bulb and blades. The decreases in $R_{0+}$ in the RE and NIR region were far greater than decreases in $R_{0+}$ observed across the visible region of the spectra (Figure 3a,b, insets). With initial submersion below the water's surface, the largest declines in the visible wavelength ranges were seen at 600 nm and 645 nm, although all peaks in the visible region continued to decrease with submersion (Figure 3a,b). While the $R_{0+}$ at the absorption feature between 400–550 nm initially decreased with submersion, the reflectance then rose as the depth of submersion increased. In the NIR region of spectra for both structures, once kelp was submerged, the broad NIR peaks were

replaced by two peaks centered around 715 nm and 815 nm (Figure 3a,b), hereafter referred to as the RE peak and the NIR peak, respectively. At each depth, the $R_{0+}$ at the RE peak was higher than the NIR peak. As submergence increased, the position of the RE peak shifted toward lower wavelengths within the RE wavelength ranges while the position of the NIR peaks remained relatively stable. The small peaks at 761 nm remained stable, but decreased in magnitude with submersion, becoming difficult to visibly distinguish around 50 cm depth (Figure 4a,b).

$R_{0+WV}$ and $R_{0+MSRE}$ showed the same general patterns as the hyperspectral data (Figure 5a–d). However, some spectral information was lost with the reduction of spectral resolution, such as the location and magnitude of different peaks. Overall, the differences in width and placement of bands resulted in only small differences in $R_{0+WV3}$ and $R_{0+MSRE}$ band values (Figure A1a,b). For both bulbs and blades at the surface, differences in the visible wavelength ranges between $R_{0+WV}$ and $R_{0+MSRE}$ were less than 0.8% for the red, blue, and green bands, and these differences became even smaller as kelp was submerged. In the RE and NIR bands, differences between $R_{0+WV}$ and $R_{0+MSRE}$ were less than 0.3% on the surface. Once submerged to 10 cm, differences between $R_{0+WV}$ and $R_{0+MSRE}$ increased to 1.8% in the NIR bands and 0.5% in the RE bands, although similar to the visible bands, the differences between $R_{0+WV}$ and $R_{0+MSRE}$ also became smaller as the kelp was submerged deeper.

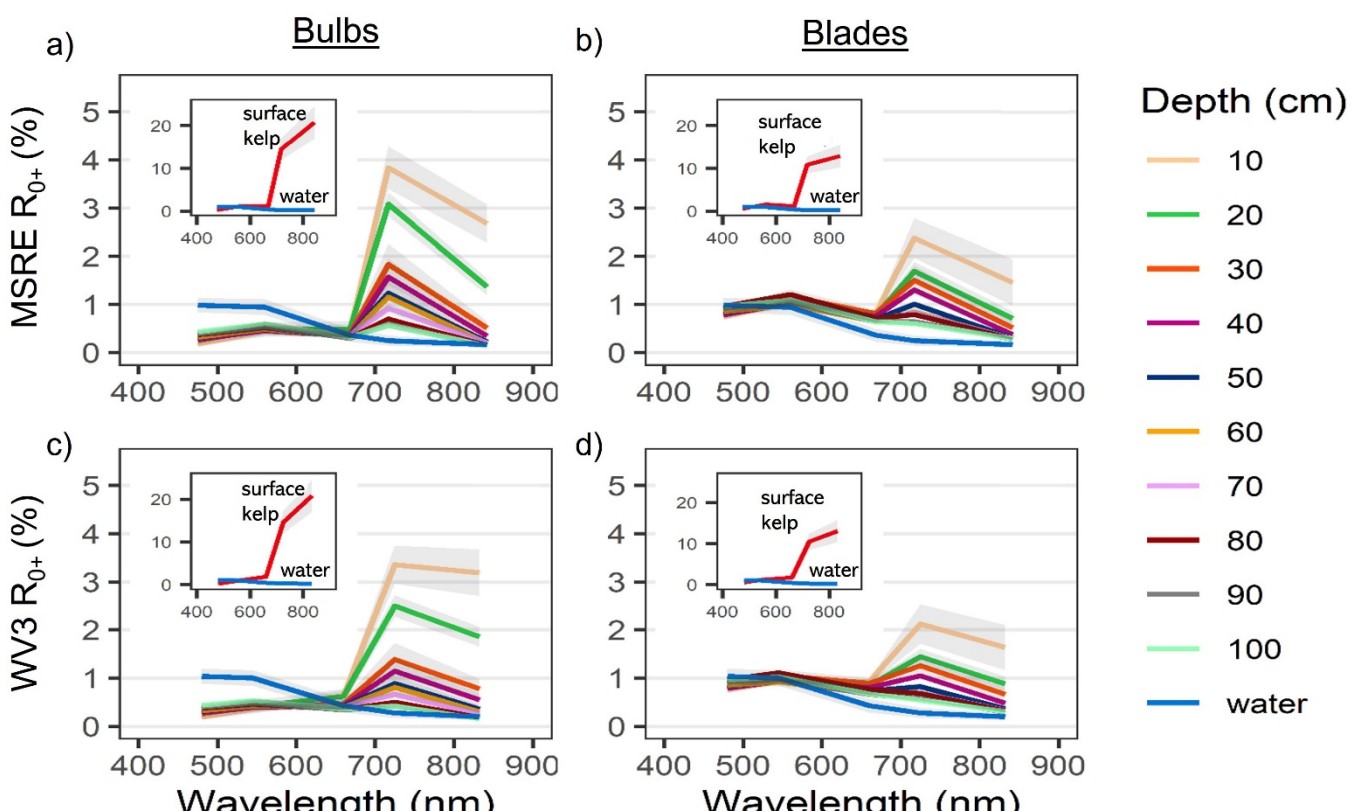

**Figure 5.** Reflectance values (R0+) for bulbs (**a**,**c**) and blades (**b**,**d**) of simulated bands (mean +/− sd) shared by the Micasense RedEdge-MX (MSRE; **a**,**b**) and WorldView-3 (WV3; **c**,**d**), derived from the hyperspectral data (Figure 3) using Gaussian response functions (Figure 2).

### 3.2. Vegetation Indices: Signal Strength and Depth-Detection Limits of Submerged Kelp

Generally, RE $VI_n$ values were higher than NIR $VI_n$ values at a given depth as kelp was submerged (Figure 6). For bulbs, RE $VI_n$ values decreased linearly from the surface to 100 cm, while NIR $VI_n$ showed a steeper linear decrease over the first 50 cm, followed by an inflection point and a lesser decline towards 100 cm. For blades, trendlines of both NIR

and RE VI$_n$ resemble exponential functions, with the NIR VI$_n$ displaying a steeper decrease of values than the RE VI$_n$.

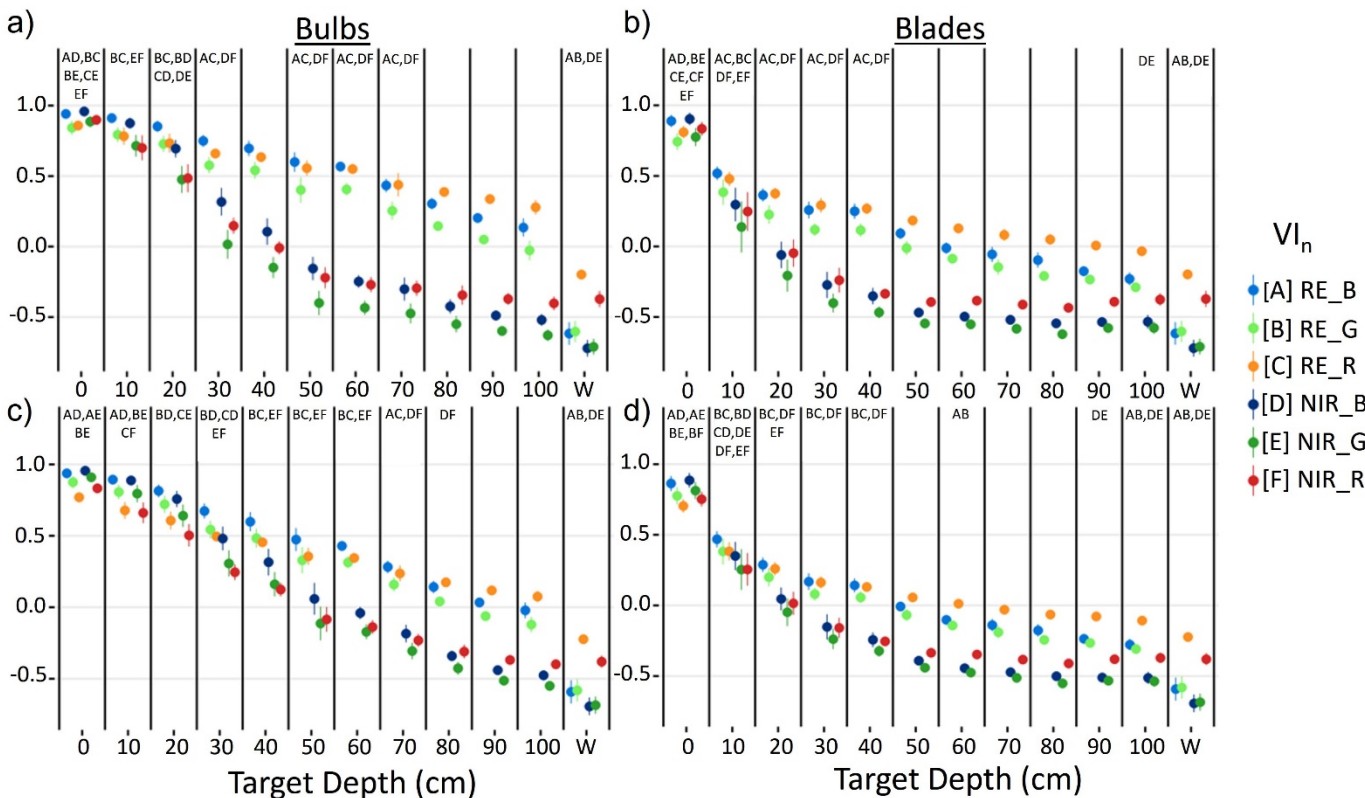

**Figure 6.** Mean +/− sd of vegetation index (VI$_n$) values for Nereocystis bulbs (**a**,**c**) and blades (**b**,**d**), submerged from the surface to 100 cm and water; derived from simulated Micasense RedEdge-MX (MSRE; **a**,**b**) and WorldView-3 (WV3; **c**,**d**) bandwidths. Paired letters above each column represent no significant differences ($p \geq 0.05$) between mean index values at that depth.

Specifically, the Games–Howell post hoc tests showed that for kelp at the surface, RE VI$_n$ values were either smaller than or not significantly different from their counterpart NIR indices (Figure 6), depending on the visible band used. Once kelp was submerged, RE VI$_n$ values were significantly greater than their NIR counterparts at each depth with the MSRE sensor. However, with the WV3 sensor, RE VI$_n$ values were not significantly greater than their NIR counterparts until 10 cm and 20 cm depth for blades and bulbs, respectively. All VI$_n$ values for water were negative, meaning that the $R_{0+}$ at the visible band used in the VI$_n$ was higher than the $R_{0+}$ at the RE or NIR band used in the VI$_n$, regardless of sensor simulation or index combination. RE_R consistently showed the highest values for water, followed by NIR_R, and there were no significant differences between RE_B and RE_G water values, nor for NIR_B and NIR_G water values. Here, we focused on the statistical results comparing the RE and NIR counterpart indices only (e.g., NIR_R & RE_R, or NIR_B & RE_B) at each depth, however, Figure 6 displays paired letters to indicate all pairs of VIn where no significant difference between VIn pairs was detected.

The depth detection limits varied based on sensor type, kelp structure, and thresholding method (Table 5; Figure 7). Overall, when using the conservative (more realistic) threshold of zero, RE VI$_n$ showed detection of kelp at least twice as deep as NIR VI$_n$, and bulbs were detectable at greater depths than blades. Detection limits for the same VI$_n$ between sensors were generally within a range of 0–20 cm, although in a few cases (e.g., RE_R) these differences were larger. In addition, the choice of different visible bands for a VI$_n$ only resulted in detection limit differences up to 20 cm, with RE_R once again proving the exception. No RE indices crossed below the dynamic thresholds at 100 cm

depth, meaning RE indices could detect kelp to at least 100 cm depth with these thresholds, while NIR indices could generally detect kelp to around 100 cm depth or less. In all cases, the RE indices at 100 cm depth were more separable from water than the NIR indices at the same depth. The use of different visible bands in the $VI_n$ combination generally resulted in detection limit differences of 0–30 cm for bulbs. For all measured depth detection limits, the index values measured at the increments 10 cm above and below the threshold remained divergent ($p < 0.05$), suggesting that all the measured results for conservative and dynamic thresholds are accurate to at least 10 cm increments.

**Table 5.** Depth detection limits (cm) based on conservative threshold of 0.0 and the dynamic thresholds (maximum water value) for Nereocystis bulbs and blades, as simulated to Micasense RedEdge-MX (MSRE) and WorldView-3 (WV3) bandwidths.

|  | Index |  | RE_B | RE_G | RE_R | NIR_B | NIR_G | NIR_R |
|---|---|---|---|---|---|---|---|---|
| **MSRE** | **Bulb** | **Conservative (0.0)** | >100 | 90 | >100 | 40 | 30 | 30 |
|  |  | **Dynamic (max.)** | >100 | >100 | >100 | >100 | 90 | 50 |
|  | **Blade** | **Conservative (0.0)** | 50 | 40 | 90 | 10 | 10 | 10 |
|  |  | **Dynamic (max.)** | >100 | >100 | >100 | >100 | >100 | 30 |
| **WV3** | **Bulb** | **Conservative (0.0)** | 90 | 80 | 100 | 50 | 40 | 40 |
|  |  | **Dynamic (max.)** | >100 | >100 | >100 | >100 | >100 | 80 |
|  | **Blade** | **Conservative (0.0)** | 40 | 40 | 60 | 20 | 10 | 20 |
|  |  | **Dynamic (max.)** | >100 | >100 | >100 | >100 | >100 | 40 |

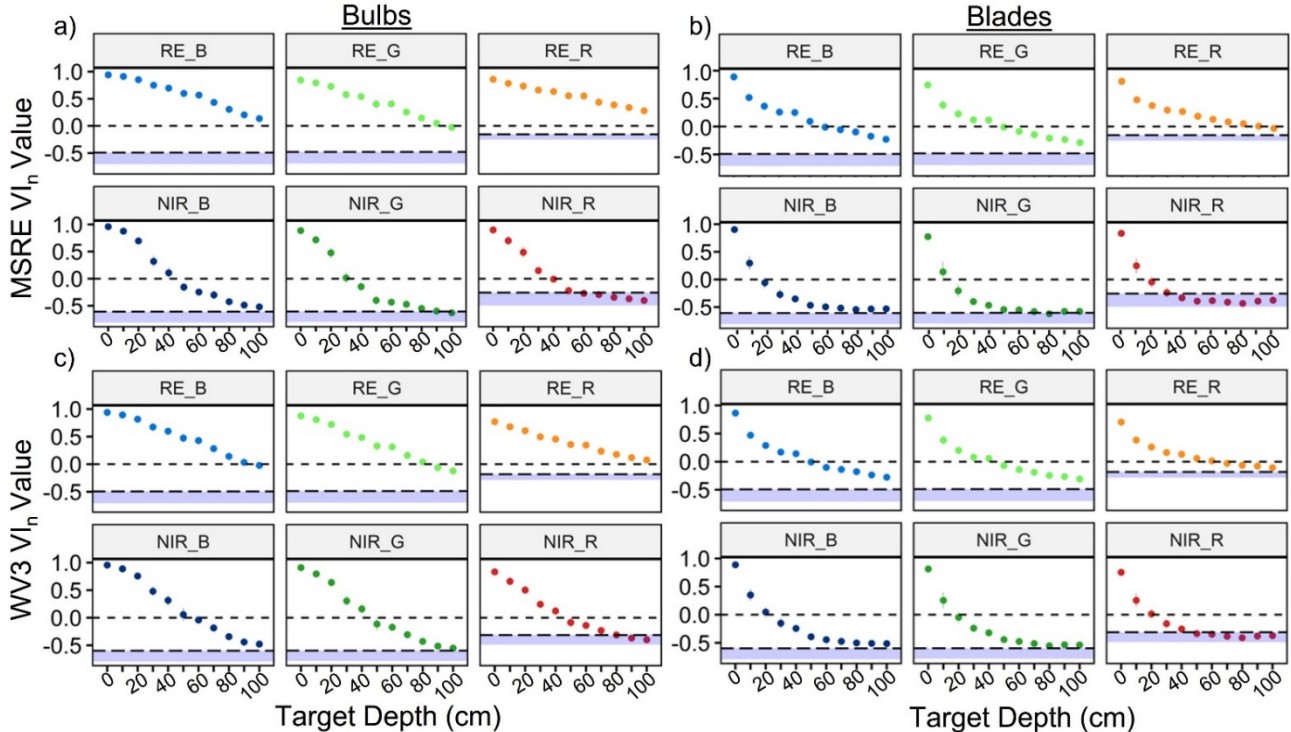

**Figure 7.** Mean +/− sd of vegetation index ($VI_n$) values for Nereocystis bulbs (**a,c**) and blades (**b,d**) submerged from the surface to 100 cm. derived from simulated Micasense RedEdge-MX (MSRE; **a,b**) and WorldView-3 (WV3; **c,d**) bandwidths. The black dashed lines at 0 represent the more conservative and realistic threshold, and the blue bars represent the full range of water values for each respective index, with the adjacent dashed lines representing the dynamic threshold.

## 4. Discussion

Overall, we found that submersion of kelp in water changes the shape and magnitude of $R_{0+}$ in the RE and NIR region of kelp spectra (Figure 3), and *Nereocystis* bulbs had a higher magnitude $R_{0+}$ in the RE and NIR region than blades (Figure 3). We also observed that RE $VI_n$ values for submerged kelp had higher separability from water than their NIR counterparts (Figure 6), meaning that kelp can be positively classified at deeper depths when using an RE $VI_n$ (Table 3; Figure 7). Our results also showed that $VI_n$ that used a visible band with high $R_{0+}$ (e.g., green or blue) had worse detectability for submerged kelp than a $VI_n$ that used a visible band with low $R_{0+}$ (e.g., red). Together, these findings have important implications for the application of kelp remote sensing to the applied monitoring of kelp forests.

### 4.1. Spectral Characteristics of Kelp as It Is Submerged

A broad $R_{0+}$ peak across the NIR region was observed for surface measurements of *Nereocystis* as a result of the interaction of light with the cellular structure of the kelp [51]. Once submerged, our experiment showed two key changes in the NIR region of the kelp spectra (Figure 3), both due to characteristic absorption features of water: (1) the splitting of the single broad NIR peak into two narrower RE and NIR peaks due to prominent water absorption feature at 760 nm [52] and (2) higher $R_{0+}$ at the RE peak versus the NIR peak, resulting from the continually increasing absorption of light by water above 600 nm [22]. Since *Macrocystis* and *Nereocystis* are spectrally similar to one another in the NIR region [15,18,27,50], and the changes seen in the spectra of submerged kelp are due to properties of water absorption, we expect that the spectral results of this experiment are generally applicable to both *Macrocystis* and *Nereocystis* canopies, making these findings relevant for surface-canopy forming kelp species globally.

The results of these experiments were generally in line with our expectations according to similar studies of submerged aquatic vegetation [24,25], although there were some interesting phenomena seen in the spectra that are worth noting. In the visible region of the spectra, $R_{0+}$ in the red wavelength range decreased with depth, as expected. However, the $R_{0+}$ at the absorption feature between 400 and 550 nm increased slightly with submersion. We hypothesize that this increase in $R_{0+}$ is due to the scattering of light by the conditions of the water optical constituents, thus increasing the $R_{0+}$ with depth. As such, we suspect that this increase in $R_{0+}$ may be specific to the water conditions during the experiment and may not have occurred if the water had contained more optical constituents that absorb blue light, such as colored dissolved organic matter. Another interesting phenomenon noted in the floating kelp spectra was what appeared to be a sunlight-induced chlorophyll fluorescence (SICF) peak at 761 nm (Figure 4). Within the NIR region, photosynthetic organisms generally have a broad SICF peak centered at 740 nm [53]. However, due to the high magnitude of the NIR reflectance, the SICF is usually only visible as a small, narrow peak centered at 761 nm. Typically, the $R_{0+}$ within the NIR wavelength range overwhelms the signal from SICF, however, atmospheric gasses highly absorb incoming irradiance at 761 nm, which can create a fill-in effect by the SICF in this region [53]. While this phenomenon has been correlated with photosynthetic output and general health of terrestrial vegetation and phytoplankton [53,54], we are not aware of any publications that report an SICF peak in kelp spectra, and this may present an opportunity for future hyperspectral research. Once kelp was submerged the SICF feature was dampened, and therefore future research should take note of the amount of kelp at the surface if attempting to derive information from an SICF peak.

### 4.2. NIR Differences between Nereocystis Bulbs and Blades

The magnitude of reflectance across the NIR region in vegetation is generally due to the cellular structure of the respective tissues [55]. Both *Nereocystis* bulb and blade tissues are composed of the same three cellular layers: the meristoderm, the medulla, and the cortex. The meristoderm is a thin chloroplast-packed epidermal layer that surrounds the entire

individual [56], and the medulla is a complex web of filaments that acts as a transportation system within the kelp, composing the innermost layer of kelp tissue [57,58]. Between these two layers is the cortex, which connects the meristoderm to the medulla, and generally provides structural support for the kelp [56,59]. Given this structural arrangement, we speculate that the NIR signal from bulbs is consistently higher compared to the blades' signal because (1) the bulb cortex is many times thicker than the blade cortex [57,59]; and (2) the gas cavity of the bulb is lined by the medulla [57], creating a high surface area with many large refractive differences—similar to the mesophyll layer of a terrestrial leaf [55,60]. In comparison, the blade medulla is housed in a gelatinous extracellular matrix between cells [57], and with no gas cavities, the refractive differences are much smaller, allowing for increased transmittance of NIR light through the blades [55]. For our experiment, spectral measurements for blades were taken using a single blade wrapped around the polyethylene frame with only slight overlap between the edges of the blade. However, *Nereocystis* individuals may have between 30–60 blades each. Overall, a thicker mass of blade tissues due to high overlap may result in higher $R_{0+}$ in the RE and NIR wavelength ranges than seen in this experiment.

### 4.3. The Implications of $VI_n$ Saturation for Detection of Floating and Submerged Kelp

When the density or biomass of the vegetation increases within a pixel of remote sensing imagery, the $VI_n$ for that pixel will asymptotically approach a saturation (i.e., a high $VI_n$ value) [39,61]. This happens because when vegetation is dense, the $R_{0+}$ at band 2 (NIR or RE) is large relative to the $R_{0+}$ at band 1 (the visible band). Our spectral measurements contained 100% kelp within the field of view, and accordingly, the $VI_n$ values calculated from the multispectral simulations were saturated when kelp was at the surface. Therefore, it is critical to understand how saturation affected the $VI_n$ values of floating kelp, as well as when kelp was submerged. For example, our WV3 simulations for bulbs at the surface showed that the $R_{0+}$ at NIR and RE bands were large compared to the red band (21%, 14%, and 1%, respectively). As such, both NIR_R and RE_R indices for bulbs at the surface were approaching saturation (0.83 and 0.77 respectively) and either index would perform relatively well for detecting floating kelp if a $VI_n$ of zero was used as a threshold to classify kelp and water. When the bulb was submerged, the $R_{0+}$ in the NIR and RE bands decreased rapidly by 10 cm depth (3.2 and 3.4% respectively) but were still relatively high compared to the red band, which had also decreased (0.6%), and therefore the NIR_R and RE_R values (0.66 and 0.68 respectively) were still relatively saturated, despite the large decreases in $R_{0+}$ at the RE and NIR bands (Figure 5). As the kelp continued to be submerged, the $R_{0+}$ at the NIR, RE, and red bands all continued to decrease, however, the $R_{0+}$ at the NIR band decreased at a faster rate and therefore the NIR_R value dropped below the threshold of zero by 50 cm while the RE_R value was still above the threshold by 100 cm. Ultimately, this example shows that due to $VI_n$ saturation, the choice of RE or NIR will make little to no difference in classification of kelp at or near the surface. However, once submerged, the use of an RE $VI_n$ will still detect kelp deeper than an NIR $VI_n$.

### 4.4. Depth Detection Limits and Separability between Kelp and Water

While it is important to understand how $VI_n$ values change as kelp is submerged, ultimately the accuracy of submerged kelp classification depends on the spectral separability between the submerged kelp and water. Here, we defined the depth at which kelp and water were no longer separable as the depth at where $VI_n$ values for submerged kelp decreased below the threshold value. RE $VI_n$ values for kelp and water had higher separability at deeper depths than their NIR counterparts (Figure 6), meaning that deeper kelp can be accurately classified when using an RE $VI_n$. Higher separability between kelp and water classes when using RE $VI_n$ has been documented using both high spatial-resolution multispectral UAV imagery [13] and with moderate spatial-resolution multispectral satellite imagery of *Macrocystis* [42], indicating that slight submergence of kelp surface-canopy may play a larger role in detection than previously thought.

While the choice between RE and NIR $VI_n$ was an important factor in submerged kelp detection, the choice of the visible band can also shift the detection limits of submerged kelp. Our results show that both the water and submerged kelp spectra had higher $R_{0+}$ in the green and blue wavelength ranges than in the red, and as such, submerged kelp became undetectable at shallower depths when using NIR_R compared to NIR_G or NIR_B. In the visible wavelength ranges, red is absorbed fastest by the water column, and in our experiment, the NIR signal is generally absorbed by around 50 cm depth, making it reasonable for this pairing to consistently have the shallowest detection limits for submerged kelp. At depths where the RE or NIR signal of kelp can no longer be detected, Figure 7 shows that subtle differences between kelp and water in the blue and green bands can still result in the kelp signal remaining above the dynamic threshold. However, these differences are small, and because conditions during the experiment were controlled, the added spectral noise from in situ environmental factors would likely complicate the detection of both surface and submerged kelp in more realistic situations. For example, the blue wavelength ranges can be highly compromised in remote sensing imagery [29,46], with local variation in atmospheric composition reducing the certainty of accuracy for blue band values. Additionally, the optical constituents of coastal water can be highly spatiotemporally variable—affecting all regions of the spectra [28,46]. At high concentrations phytoplankton in the water column may result in changes to reflectance in the visible wavelength ranges as well as high RE or NIR reflectance [62], while changes to optical constituents such as sediment or CDOM may also impede the detection of submerged kelp [47,63].

In this experiment, the optical water conditions (Secchi = 7.5) were typical of the coastal waters of British Columbia [18,28,64]. Considering the Secchi measurement, the local depth (12 m), and the $R_{0+}$ from water with no kelp (Figure 3), the bottom substrate signal was not part of the measured $R_{0+}$ in our experiment. Yet kelp on the coast of British Columbia is often found as fringing canopies near the shoreline [18], which can result in a strong contribution of benthic substrate to the $R_{0+}$ measured by space and air-borne platforms. Reflectance from shallow benthic features can result in highly variable $R_{0+}$ in both the visible and near-infrared wavelength ranges, resulting in misclassification of submerged vegetation as canopy kelp [47,64]. Therefore, it is important to understand site characteristics (e.g., bathymetry and water turbidity) to define better the use of NIR or RE for kelp classification. For instance, if enough understanding of the local conditions at the time of imagery acquisition is not available, it may be more appropriate to use NIR_R to reduce the addition of signal of the bottom substrate. Alternately, if imagery or associated ground truth data have a high enough spatial resolution (e.g., from UAV or other aerial platforms), visual interpretation of surface-canopy morphology from expert knowledge may be adequate for manual classification or ground truthing when using an RE $VI_n$.

### 4.5. Implications for Mixed Pixels

During the experiment, spectral data were collected using a small footprint to reduce uncertainties associated with having the reflectance signal of multiple targets within the field of view (i.e., mixed pixels). However, remote sensing imagery often contains mixed pixels [16,35]. This becomes especially problematic when sensors have a lower spatial resolution, where erroneous classification of a pixel as kelp may result in the overestimation of total kelp canopy. Multiple end-member spectral mixture analysis (MESMA) is an approach that has been applied to satellite imagery for both *Macrocystis* [11,35] and *Nereocystis* canopy [19,65] to determine what proportion of the pixel is kelp, and what proportion is water. When MESMA is applied to remote sensing imagery for kelp detection, it is assumed that all $VI_n$ or band values within a pixel are a linear combination of kelp and water end-members [35]. However, if the kelp fraction within a pixel is low enough, the spectral contribution from water may overwhelm the kelp signal, lowering the overall pixel value and allowing the pixel to be erroneously classified as water [19,27]. Our results suggest that if submerged kelp is present when MESMA is performed, which is most often the case, the

reduced signal from the submerged kelp within the pixel may lead to an underestimation of the kelp fraction within the pixel. Using an RE VI$_n$ when performing MESMA may allow the user to detect more submerged kelp, thus contributing to a higher overall pixel value and increasing the accuracy of the classification. This may be especially relevant if attempting to determine relationships between remote sensing imagery and biomass, since *Nereocystis* blades show a higher correlation to the mass of the individual than any other metric tested [10]. Further, *Nereocystis* canopy generally has less dense biomass at the surface than *Macrocystis* [65,66], and, therefore, is more likely to be misclassified in moderate or low spatial resolution imagery.

## 5. Conclusions

Our experiment contributes new, detailed information on the effects of kelp submersion on the above water reflectance, as well as a comparison of the depth detection limits of kelp when using red-edge and near-infrared indices. We determined that the near-infrared region of kelp spectra is strongly absorbed upon submersion, however, there is a narrow spectral peak in the red-edge region that can be used to enhance the remote sensor's ability to detect submerged kelp due to lower water absorption. Detection limits varied based on kelp tissue, the thresholding method, and the visible band used in the vegetation index calculation, but overall, red-edge vegetation indices detected deeper than their counterpart NIR indices, which may allow the remote sensor to improve accuracy when mapping sparse and partially submerged kelp canopy or attempting to derive biomass from canopy reflectance values. Kelp forests may be mapped using remote sensing for various reasons, ranging from estimation of biomass for kelp harvesting to multi-year temporal analyses to assess the impacts of environmental drivers on kelp ecosystems. Yet kelp systems can be highly variable in abundance between years, and our study shows that the spectral variables used to detect kelp canopy in remote sensing imagery play an important role in the amount of submerged kelp canopy detected. Therefore, it is critical for a remote sensing user to understand how the physical interaction between light and water may affect the depth at which kelp can be detected. For example, RE VIn might be especially useful if resource managers are attempting to set quotas for harvestable biomass of *Nereocystis* and wish to detect as much blade biomass as possible for specific beds. However, if one wishes to reduce detection of subsurface kelp canopy or other shallow benthic vegetation, we recommend the use of the NIR_R (NDVI), which consistently had the shallowest detection limits of the indices tested.

**Author Contributions:** B.T., M.C. and L.Y.R. designed the study. B.T. and L.Y.R. carried out the experiment. B.T. conducted the analysis and wrote the manuscript with input and guidance from M.C., L.Y.R., F.J. and M.H.-L. All authors have read and agreed to the published version of the manuscript.

**Funding:** During this research BT was supported through a MITACS Accelerate internship with the Hakai Institute, as well as an NSERC CGS-M award and Costa's NSERC-DG.

**Institutional Review Board Statement:** Not applicable.

**Informed Consent Statement:** Not applicable.

**Data Availability Statement:** Data are available for research purposes upon request to the authors' institutions.

**Acknowledgments:** We thank the Hakai Institute and the Canada NSERC-DG for providing funding for this research as well as Robert Atwood for letting us use his slip at the Oak Bay marina to test this experiment. Also, thank you to Lianna Gendall for helping with the kelp in the graphical abstract.

**Conflicts of Interest:** The authors declare no conflict of interest.

**Appendix A**

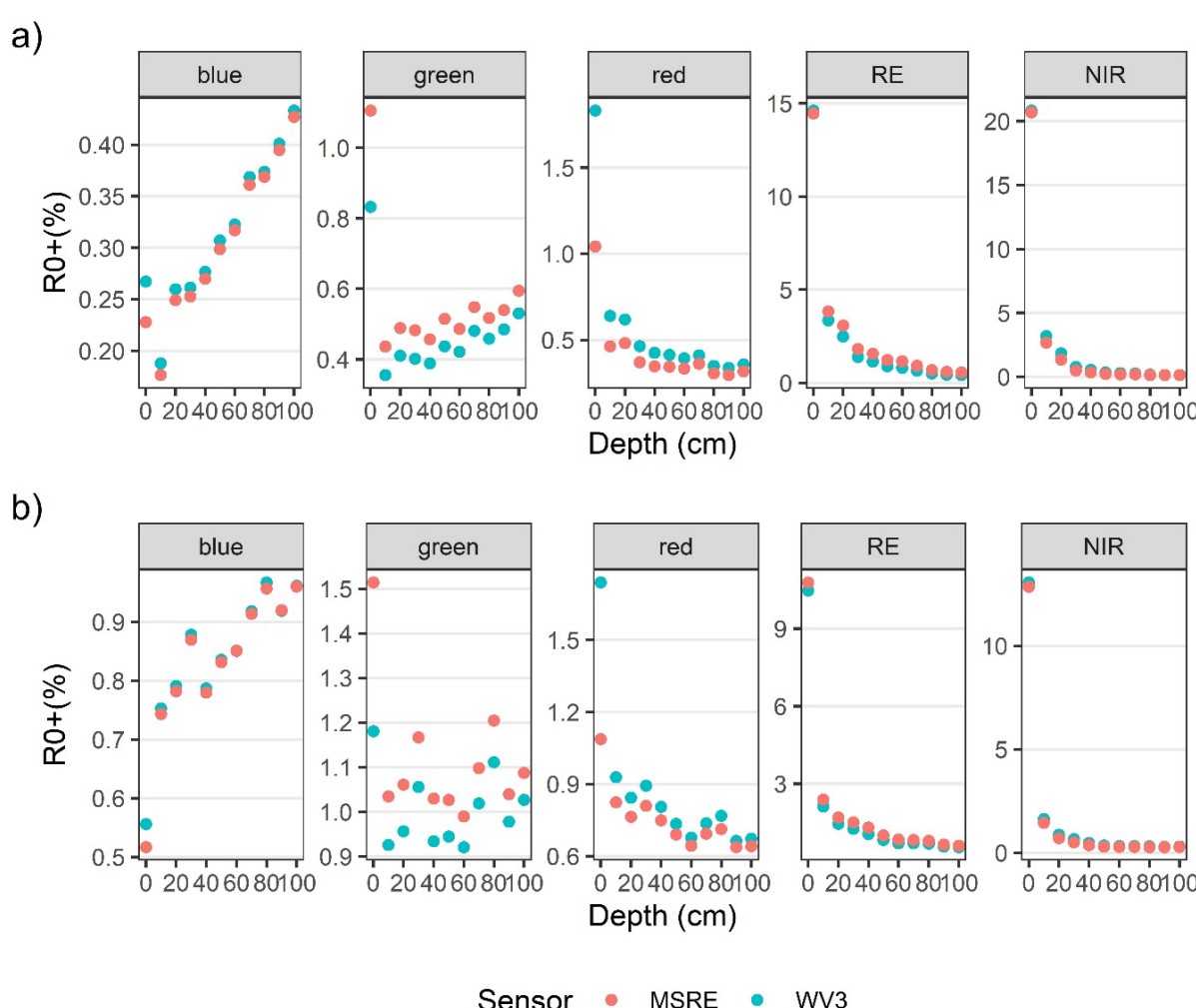

**Figure A1.** Differences between $R_{0+}$ values of each shared band of the Micasense RedEdge-MX (MSRE) and WorldView-3 sensors, as simulated from the bulb (**a**) and blade (**b**) spectral measurements.

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
