# Peer review of "Comparing the Use of Red-Edge and Near-Infrared Wavelength Ranges for Detecting Submerged Kelp Canopy"

_remotesensing, doi:10.3390/rs14092241_

Round 1

Reviewer 1 Report

The authors carried out an experiment to compare the depth detection limits of kelp when using red-edge and NIR indices. It showed that a narrow spectral peak in the red-edge region can be used to detect submerged kelp due to lower water absorption. Red-edge indices detected deeper than their counterpart NIR indices. It's a meaningful work to provide a potential way for remote sensor such as WorldView-3 multispectral satellite sensor to  improve accuracy when mapping sparse and partially submerged kelp canopy.

English expression needs minor revision.

Reviewer 2 Report

The work interested me and has potential, especially when using modern methods that are support this project. Due to the many expressions in the spectroscopic part, the wording in some sentences is confusing, not clear. It is worth taking care of it and improving it in the article. Hence the quality of writing needs to be improved:

In title authors use incorrect “…Near-infrared Wavelengths”. It should be “…Near-infrared wavelength ranges”

In abstract correct: near-infrared signal use: near-infrared spectral region or near-infrared spectral range.

Line 58: “…infrared wavelengths (NIR) (700-1000 nm),…” instead of wavelengths it should be wavelength range. Please check this article and make corrections wherever possible.

Line 80: “…kelp canopy affects the spectral signal in the NIR region...” this part of the text is incomprehensible. "Spectral signal" is not correct, it should be "spectral region".

Line 83: “…shorter NIR wavelengths in the red-edge (RE) region of the spectrum…” I think you should spectra or spectral regions.

Line 94: “…[11,17–20,27]…”  This way of citation is confusing please correct it.

Line 97: “To address these knowledge gaps, we..” Please use another term.

Line 127: “…two of the four trials used the Nereocystis bulb/pneumatocyst, and two trials used…” What does it mean? Please correct it.

Line: 163: “...Each R0+ spectrum was first smoothed using a mean filter…” You should use “spectra

Line 467: “…RE and NIR peaks/bands are relatively small at depth…”. What does it mean? Please correct it.

Authors use the acronym NIR for many keywords. Please correct it everywhere and replace that: NIR (near-infrared).

The authors use the solution using spectroscopy in their research. Because the Fourier transform infrared method is also currently used, it is more precise than the Near-infrared method, why has it not been used?

In future, how will do you plan to use the output results? How will this solution affect the ecosystem?

Reviewer 3 Report

See attached file.

Reviewer 4 Report

This work presents a detailed methodology based on several experiments to reach the proposed hypothesis. Using multispectral analysis, the authors introduce a comprehensive analysis for detecting submerged kelp canopy. However, the work has some lacks and needs minor corrections.

1.- Although the hypothesis is justified with different experiments, the text is really exhausting to read, mainly because the text is saturated with values related to dimensions, angles, wavelength, etc. The authors need to find an alternative to present this relevant information. I suggest using Tables to summarize the text.

2.- On the other hand, the Abstract section is too long for scientific reports. In this type of work, it is recommendable to write three sentences for the introduction part (Background, A-state-of-the-art, Problem), three sentences for the methods part,  three sentences for relevant results, and one sentence for the conclusion. 

3.- The vast set of parameters described in the text should be summarized in a general table. In addition, the technical characteristics of the instrumentation could be added to a table.

4.- Why the value (10 cm) is between parenthesis in Lines 22 and 124?

5.- Although figures are a fast and outstanding way to present results, it is necessary that the information provided in the text is reflected in the respective figure. Some problems are related to:
(a) Line 293-296: Where is it possible to find this information? This premise is related to Fig.3
(b) Authors cite Fig.3 or Fig. 4 in the text, but both figures have two subfigures. Which one do the authors refer to?
(c) Line 296-297: The authors claim that there is a small, narrow peak centered at 761 nm; however, it is possible to see a peak at 761 nm in Fig.4a and a peak at 750 nm in Fig. 4b.

(d) Figure 8 needs a legend to understand.

6.- The authors need to correct and standardize the style of the figures. Also, several figures are distorted (font and letter size).

7.- I suggest rewriting the text considering the results' reproducibility by the scientific community. 

Round 2

Reviewer 3 Report

No more comments.